# Robust Calibration with Multi-domain Temperature Scaling

**Yaodong Yu**
University of California, Berkeley
yyu@eecs.berkeley.edu

**Stephen Bates**
University of California, Berkeley
stephenbates@cs.berkeley.edu

**Yi Ma**
University of California, Berkeley
yima@eecs.berkeley.edu

**Michael I. Jordan**
University of California, Berkeley
jordan@cs.berkeley.edu

## Abstract

Uncertainty quantification is essential for the reliable deployment of machine learning models to high-stakes application domains. Uncertainty quantification is all the more challenging when training distribution and test distribution are different, even if the distribution shifts are mild. Despite the ubiquity of distribution shifts in real-world applications, existing uncertainty quantification approaches mainly study the in-distribution setting where the train and test distributions are the same. In this paper, we develop a systematic calibration model to handle distribution shifts by leveraging data from multiple domains. Our proposed method—multi-domain temperature scaling—uses the heterogeneity in the domains to improve calibration robustness under distribution shift. Through experiments on three benchmark data sets, we find our proposed method outperforms existing methods as measured on both in-distribution and out-of-distribution test sets.

## 1   Introduction

To make learning systems reliable and fault-tolerant, predictions must be accompanied by uncertainty estimates. A significant challenge to accurately codifying uncertainty is the distribution shift that typically arises over the course of a system's deployment [Quiñonero-Candela et al., 2008]. For example, suppose health providers from 20 different hospitals employ a model to make diagnostic predictions from fMRI data. The distributions across hospitals could be quite different as a result of differing patient populations, machine conditions, and so on. In such a setting, it is critical to provide uncertainty quantification that is valid for *every* hospital—not just on average across all hospitals. Going even further, our uncertainty quantification should be informative when a new 21st hospital goes online, even if the distribution shifts from those already encountered. As another example, a centralized model is trained on training data from existing clients in federated learning. It is important for the central server to provide uncertainty quantification for every client. Similar to the fMRI example, the centralized model should still produce valid uncertainty quantification for unseen new clients. Another example is applying animal recognition models on images in wildlife monitoring, where one set of camera traps corresponds to one domain, and the model will be deployed under distribution shift, i.e., new camera traps. In this work, we study calibration in the multi-domain setting. We find that by requiring accurate calibration across all observed domains, our method provides more accurate uncertainty quantification on unseen domains.

Calibration is a core topic in learning [Platt et al., 1999; Naeini et al., 2015; Gal and Ghahramani, 2016; Lakshminarayanan et al., 2017; Guo et al., 2017; Bates et al., 2021], but most techniques are targeted at settings with no distribution shift. To see this, we consider a simple experiment on the

36th Conference on Neural Information Processing Systems (NeurIPS 2022).

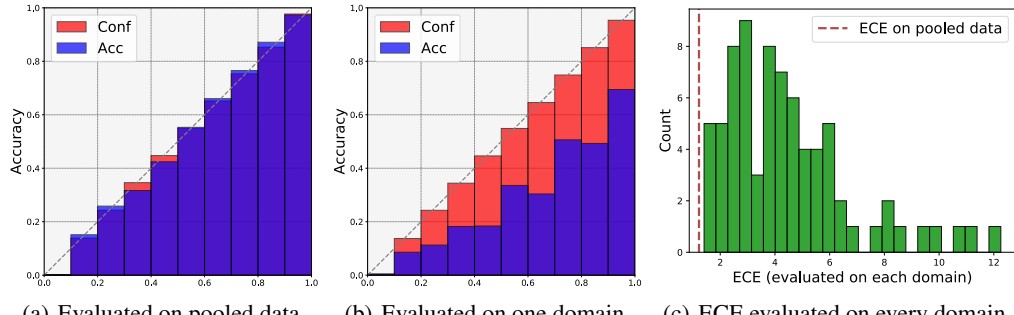

| (a) Evaluated on pooled data. | (b) Evaluated on one domain. | (c) ECE evaluated on every domain. |

Figure 1: Reliability diagrams and expected calibration error histograms for temperature scaling with a ResNet-50 on ImageNet-C. We use temperature scaling to obtain adjusted confidences for the ResNet-50 model. **(a)** Reliability diagram evaluated on the pooled data of ImageNet-C. **(b)** Reliability diagram evaluated on data from one domain (Gaussian corruption with severity 5) in ImageNet-C. **(c)** Calibration evaluated on every domain in ImageNet-C as well as the pooled ImageNet-C (measured in ECE, lower is better).

ImageNet-C [Hendrycks and Dietterich, 2019] dataset, which consists of 76 domains. Here, each domain corresponds to one type of data corruption applied with a certain severity. We apply the temperature scaling technique [Guo et al., 2017] on the pooled data from all domains. In Figure 1(a) and 1(b), we display the reliability diagrams for the pooled data and for one individual domain. We find that even under a relatively mild distribution shift—i.e., subpopulation shift from the mixture of all domains to the single domain—temperature scaling does not produce calibrated confidence estimates on the stand-alone domain. This behavior is pervasive; in Figure 1(c), we see that the calibration on individual domains is much worse than the the reliability diagram from the pooled data would suggest.

To address this issue, we develop a new algorithm, *multi-domain temperature scaling*, that leverages multi-domain structure in the data. Our algorithm takes a base model and learns a calibration function that maps each input to a different temperature parameter that is used for adjusting confidence in the base model. Empirically, we find our algorithm significantly outperforms temperature scaling on three real-world multi-domain datasets. In particular, in contrast to temperature scaling, our proposed algorithm is able to provide well-calibrated confidence on each domain. Moreover, our algorithm largely improves robustness of calibration under distribution shifts. This is expected, because if the calibration method performs well on every domain, it is likely to have learned some structure that generalizes to unseen domains. Theoretically, we analyze the multi-domain calibration problem in the regression setting, providing guidance about the conditions under which robust calibration is possible.

**Contributions.** The main contributions of our work are as follows: Algorithmically, we develop a new calibration method that generalizes the widely used temperature scaling concept from single-domain to multi-domain. The proposed new method exploits multi-domain structure in the data distribution, which enables model calibration on every domain. We conduct detailed experiments on three real-world multi-domain datasets and demonstrate that our method significantly outperforms existing calibration methods on *both in-distribution domains and unseen out-of-distribution domains*. Theoretically, we study multi-domain calibration in the regression setting and develop a theoretical understanding of robust calibration in this setting.

### Related Work

**Calibration methods.** There is a large literature on calibrating the well-trained machine learning models, including histogram binning [Zadrozny and Elkan, 2001], isotonic regression [Zadrozny and Elkan, 2002], conformal prediction Vovk et al. [2005], Platt scaling [Platt et al., 1999], and temperature scaling [Guo et al., 2017]. These calibration methods apply a validation set and post-process the model outputs. As shown in Guo et al. [2017], temperature scaling, a simple method that uses a single (temperature) parameter for rescaling the logits, performs surprisingly well on calibrating confidences for deep neural networks. We focus on this approach in our work. More broadly, there has been much recent work develop methods to improve calibration for deep learning models, including

augmentation-based training [Thulasidasan et al., 2019; Hendrycks et al., 2019b], calibration for neural machine translation [Kumar and Sarawagi, 2019], neural stochastic differential equation [Kong et al., 2020], self-supervised learning [Hendrycks et al., 2019a], ensembling [Lakshminarayanan et al., 2017], and Bayesian neural networks [Gal and Ghahramani, 2016; Gal et al., 2017], as well as statistical guarantees for calibration with black-box models Angelopoulos et al. [2021].

**Calibration under distribution shifts.** Ovadia et al. [2019] conduct an empirical study on model calibration under distribution shifts and find that models are much less calibrated under distribution shifts. Minderer et al. [2021] revisit calibration of recent state-of-the-art image classification models under distribution shifts and study the relationship between calibration and accuracy. Wald et al. [2021] study model calibration and out-of-distribution generalization. Other works consider providing uncertainty estimates under structured distribution shifts, such as covariate shift [Tibshirani et al., 2019; Park et al., 2021], label shift [Podkopaev and Ramdas, 2021], and $f$-divergence balls [Cauchois et al., 2020]. Another line of work studies calibration in the domain adaptation setting [Wang et al., 2020; Park et al., 2020], which require unlabeled samples from the target domain.

## 2 Problem setup

**Notation.** We denote the input space and the label set by $\mathcal{X} \subseteq \mathbb{R}^d$ and $\mathcal{Y} = \{1, \ldots, J\}$. We let $[x]_i$ denote the $i$-th element of vector $x$. We use $\mathcal{P}(X)$ to denote the marginal feature distribution on input space $\mathcal{X}$, $\mathcal{P}(Y|X)$ to denote the conditional distribution, and $\mathcal{P}(X, Y)$ to denote the joint distribution. For the multiple domains scenario, we let $\mathcal{P}_k(X)$ and $\mathcal{P}_k(Y|X)$ denote the feature distribution and conditional distribution for the $k$-th domain. We let $f : \mathcal{X} \to \mathbb{R}^J$ denote the base model, e.g., a deep neural network, where $J$ is the total number of classes. We assume $f$ returns an (unnormalized) vector of logits. Throughout the paper, the base model is trained with training data and will not be modified. The class prediction of model $f$ on input $x \in \mathcal{X}$ is denoted by $\hat{y} = \text{argmax}_{j \in \{1, \ldots, J\}} [f(x; \theta)]_j$. We use $\mathbf{1}\{\cdot\}$ to represent the indicator function. We use $h(\cdot; f, \beta) : \mathcal{X} \to [0, 1]$ to denote a *calibration map* (parameterized by $\beta$) that takes an input $x \in \mathcal{X}$ and returns a confidence score—this is a post-processing of the base model $f$. We let $\hat{\pi} = h(x; f, \beta) \in [0, 1]$ denote the confidence estimate for sample $x$ when using model $f$. For instance, if we have 100 predictions $\{\hat{y}_1, \ldots, \hat{y}_{100}\}$ with confidence $\hat{\pi}_1 = \cdots = \hat{\pi}_{100} = 0.7$, then the accuracy of $f$ is expected to be 70% on these 100 samples (if the confidence estimate is well calibrated). Data from the domains $\mathcal{P}_1, \ldots, \mathcal{P}_K$ are used for learning the calibration models, and we call the *in-distribution* (InD) domains. We use $\widetilde{\mathcal{P}}$ to denote the unseen *out-of-distribution* (OOD) domain which is not used for calibrating the base model. Our goal is to learn a calibration map $h$ that is well calibrated on the OOD domain $\widetilde{\mathcal{P}}$. To do this, we will learn a calibration map that does well on all InD domains simultaneously.

To measure calibration, we first review the definition of approximate expected calibration error.

**Definition 2.1** (ECE). *For a set of samples $\mathcal{D} = \{(x_i, y_i)\}_{i=1}^n$ with $(x_i, y_i) \overset{\text{i.i.d.}}{\sim} \mathcal{P}(X, Y)$, the (empirical)* expected calibration error *(ECE) with $M$ bins evaluated on $\mathcal{D}$ is defined as*

$$\text{ECE}(\mathcal{D}, M) = \sum_{m=1}^M \frac{|B_m|}{n} |\text{Acc}(B_m) - \text{Conf}(B_m)|, \tag{1}$$

*and $B_m$, $\text{acc}(B_m)$, $\text{conf}(B_m)$ are defined as*

$$B_m = \{i \in [n] : \hat{\pi}_i \in ((m-1)/M, m/M]\},$$
$$\text{Acc}(B_m) = (1/|B_m|) \sum_{i \in B_m} \mathbf{1}\{\hat{y}_i = y_i\}, \quad \text{Conf}(B_m) = (1/|B_m|) \sum_{i \in B_m} \hat{\pi}_i,$$

*where $\hat{\pi}_i$ and $\hat{y}_i$ are the confidence and predicted label of sample $x_i$.*

The empirical ECE defined in Eq. (1) approximates the expected calibration error (ECE) $\mathbb{E}[|p - \mathbb{P}(\hat{y} = y|\hat{\pi} = p)|]$ with bin size equal to $M$ Naeini et al. [2015]; Guo et al. [2017]; see Lee et al. [2022] for statistical results about about the empirical ECE as an estimator. The perfect calibrated map corresponds to the case when $\mathbb{P}(\hat{y} = y|\hat{\pi} = p) = p$ holds for all $p \in [0, 1]$.

**Multi-domain calibration.** Although the standard ECE measurement in Eq. (1) provides informative evaluations for various calibration methods in the single-domain scenario, it does not provide fine-grained evaluations when the dataset consists of multiple domains, $\mathcal{P}_1, \ldots, \mathcal{P}_K$. It is possible that the

ECE evaluated on the pooled data $\mathcal{D}_K^{\mathrm{pool}} = \mathcal{D}_1 \cup \cdots \cup \mathcal{D}_K$ is small while the ECE evaluated on one of the domains is large. For example, as shown in Figure 1(c), there may exist a domain, $k \in [K]$, such that the ECE evaluated on domain $k$ is much higher than the ECE evaluated on the pooled dataset, i.e., $\mathrm{ECE}(\mathcal{D}_k) \gg \mathrm{ECE}(\mathcal{D}_K^{\mathrm{pool}})$. In the fMRI application mentioned in Section 1, producing well-calibrated confidence on data from every hospital is a more desirable property compared to only being calibrated on the pooled data from all hospitals. Therefore, it is natural to consider the ECE evaluated on every domain, which we refer to as "per-domain ECE." Next, we introduce the notion of Multi-domain ECE to formalize per-domain calibration.

**Definition 2.2** (Multi-domain ECE). *For a dataset $\mathcal{D}_K^{\mathrm{pool}} = \mathcal{D}_1 \cup \cdots \cup \mathcal{D}_K$ consisting of samples from $K$ domains, where $\mathcal{D}_k = \{(x_{i,k}, y_{i,k})\}_{i=1}^{n_k}$ and $(x_{i,k}, y_{i,k}) \overset{\mathrm{i.i.d.}}{\sim} \mathcal{P}_k(X, Y)$, the (empirical) multi-domain expected calibration error (Multi-domain ECE) with $M$ bins evaluated on $\mathcal{D}_K^{\mathrm{pool}}$ is defined as $\mathsf{MDECE}(\mathcal{D}_K^{\mathrm{pool}}) = \frac{1}{K} \sum_{k=1}^{K} \mathsf{ECE}(\mathcal{D}_k)$.*

**Remark 2.3.** *In Definition 2.2, we weight each domain equally to balance across domains, which could better reflect how the calibration method performs on each individual domain. Furthermore, in our experiments, we also visualize the ECE measured on each domain to provide additional information on model performance on every domain.*

Compared with the standard ECE evaluated on the pooled dataset, multi-domain ECE provides information about per-domain model calibration. In the multi-domain setting, we aim to learn a calibration map $\hat{h}$ that can produce calibrated confidence estimates on every InD domain. Intuitively, if the unseen OOD domain $\widetilde{\mathcal{D}}$ is similar to one or multiple InD domains, $\hat{h}$ can still provide reliable confidence estimates on the new domain. We formally study the connection between "well-calibrated on each InD domain" and "robust calibration on the OOD domain" in Section 5.

**Temperature scaling.** Next, we review a simple and effective calibration method, named temperature scaling (TS) [Platt et al., 1999; Guo et al., 2017], that is widely used in single-domain model calibration. Temperature scaling applies a single parameter $T > 0$ and produces the confidence prediction for the base model $f$ as

$$h^{\mathrm{ts}}(x; f, T) = \max_{j \in \{1, \ldots, J\}} [\mathrm{Softmax}(f(x)/T)]_j,$$

where $[\mathrm{Softmax}(z)]_j = \exp([z]_j) / \sum_{i=1}^{J} \exp([z]_i)$. The parameter $T$ is the so-called *temperature*, with larger temperature yielding more diffuse probability estimates. To learn the temperature parameter $T$ from dataset $\mathcal{D} = \{(x_i, y_i)\}_{i=1}^{n}$, Guo et al. [2017] propose to find $T$ by solving the following convex optimization problem,

$$\min_T \mathcal{L}_{\mathsf{TS}}(T) := -\sum_{i=1}^{n} \sum_{j=1}^{J} \mathbf{1}\{y_i = j\} \cdot \log([\mathrm{Softmax}(f(x_i)/T)]_j), \qquad (2)$$

which optimizes the temperature parameter such that the negative log likelihood is minimized. We use `TS-Alg` to denote the temperature scaling learning algorithm; given inputs dataset $\mathcal{D}$ and base model $f$, `TS-Alg` outputs the learned temeperature parameter by solving Eq. (2), e.g., $\hat{T} = \mathtt{TS\text{-}Alg}(\mathcal{D}, f)$.

## 3 Multi-domain temperature scaling

We propose our algorithm—multi-domain temperature scaling—that aims to improve the calibration on each domain. One key observation is that if we apply temperature scaling to each domain separately, then TS is able to produce calibrated confidence on every domain. Therefore, the question becomes how to "aggregate" these temperature scaling models and learn one calibration model, denoted by $\hat{h}$, that has similar performance to the $k$-th calibration model $\hat{h}_k$ evaluated on domain $k$ for every $k \in [K]$.

At a high level, we propose to learn a calibration model that maps samples from the input space $\mathcal{X}$ to the temperature space $\mathbb{R}_+$. To start with, we learn the temperature parameter $\hat{T}_k$ for the base model on every domain $k$ by applying temperature scaling on $\mathcal{D}_k$. Next, we apply the base deep model to compute feature embeddings of samples from different domains,[1] and label feature embeddings from

---

[1] We use the penultimate layer outputs of model $f$ as the feature embeddings by default.

the $k$-th domain with $\hat{T}_k$. In particular, we construct $K$ new datasets, $\hat{\mathcal{D}}_1, \ldots, \hat{\mathcal{D}}_K$, where each dataset contains feature embeddings and temperature labels from one domain, i.e., $\hat{\mathcal{D}}_k = \{(\Psi(x_{i,k}), \hat{T}_k)\}_{i=1}^{n_k}$. Finally, we apply linear regression on these labeled datasets. In detail, our algorithm is as follows:

---

1. **Learn temperature scaling model for each domain.** For every domain $k$, we learn temperature $\hat{T}_k$ by applying temperature scaling on validation data $\mathcal{D}_k = \{(x_{i,k}, y_{i,k})\}_{i=1}^{n_k}$ from $k$-th domain, i.e., $\hat{T}_k = \texttt{TS-Alg}(\mathcal{D}_k, f)$ and $\texttt{TS-Alg}$ denotes the TS algorithm.

2. **Learn linear regression of temperatures.** Extract the feature embeddings of the base deep model $f$ on each domain. Use $\Psi(x_{i,k}) \in \mathbb{R}^p$ to denote the feature embedding of the $i$-th sample from $k$-th domain. Then we learn $\hat{\theta}$ by solving the following optimization problem,

$$\hat{\theta} = \operatorname*{argmin}_{\theta} \sum_{k=1}^{K} \sum_{i=1}^{n_k} \left( \langle \Psi(x_{i,k}), \theta \rangle - \hat{T}_k \right)^2 .$$

3. **Predict temperature on unseen test samples.** Given an unseen test sample $\widetilde{x}$, we first compute the predicted temperature $\widetilde{T}$ using the learned linear model $\widetilde{T} = \langle \Psi(\widetilde{x}), \hat{\theta} \rangle$. Then we output the confidence estimate for sample $\widetilde{x}$ as

$$\widetilde{\pi} = \max_{j} \left[ \operatorname{Softmax}\left( f(\widetilde{x})/\widetilde{T} \right) \right]_j .$$

---

We denote our proposed method by MD-TS (**M**ult-**D**omain **T**emperature **S**caling). A presentation of the algorithm in pseudocode can be found in Algorithm 1, Appendix A.

We pause to consider the basic concept in more detail. The goal of our proposed algorithm is to predict the best temperature for samples from different several domains. In an ideal setting where the learned linear model $\hat{\theta}$ results in good calibration on *every* InD domain, we can expect that $\hat{\theta}$ will continue to yield good calibration on the OOD domain $\widetilde{\mathcal{P}}$ when $\widetilde{\mathcal{P}}$ is close to one or several InD domains. For example, $\widetilde{\mathcal{P}}$ will work well if $\widetilde{\mathcal{P}}$ is a mixture of the $K$ domains, i.e., $\widetilde{\mathcal{P}} = \sum_{k=1}^{K} \alpha_k \mathcal{P}_k$ and $\alpha \in \Delta^{K-1}$. Regarding the algorithmic design, linear regression is one of the simplest models for solving the regression problem. It is computationally fast to learn such linear models as well as make predictions on new samples, making it attractive. We test alternative, more flexible, regression algorithms in Section 4 but do not observe significant gains over linear regression.

To illustrate how our proposed algorithm MD-TS performs differently from standard TS, we return to the ImageNet-C dataset. We compare the predicted temperature of our algorithm on new samples from domain $k$ with the temperature that results from running TS on domain $k$ alone. The results are summarized in Figure 2, where each circle corresponds to the mean predicted temperature on one InD domain. For each domain, we also visualize the standard deviation of the predicted temperatures for samples from that domain (the horizontal bar around each point). We find that our algorithm predicts the temperature quite well. Note that it does not have access to the domain index information of the fresh samples. By contrast, TS always uses the same temperature, regardless of the input point.

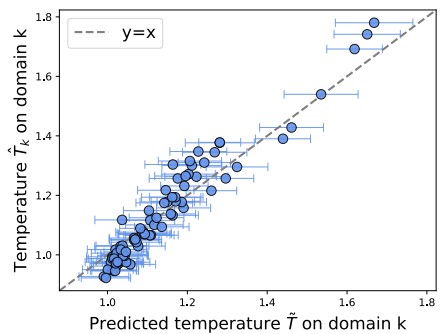

Figure 2: Compare the predicted temperature to the learned temperature $\hat{T}_k$ on the $k$-th domain.

## 4 Experiments

In this section, we present experimental results evaluating our proposed method, demonstrating its effectiveness on both in-distribution and out-of-distribution calibration. We focus on three real-world datasets, including ImageNet-C [Hendrycks and Dietterich, 2019]—a widely used robustness

Table 1: Per-domain ECE (%) comparison on three datasets. We evaluate the per-domain ECE on InD and OOD domains. We report the mean and standard error of per-domain ECE on one dataset. Lower ECE means better performance.

| Datasets | Architectures | InD-domains | | | OOD-domains | | |
|---|---|---|---|---|---|---|---|
| | | MSP | TS | MD-TS | MSP | TS | MD-TS |
| ImageNet-C | ResNet-50 | 7.36±0.28 | 5.80±0.10 | 3.84±0.05 | 6.87±0.16 | 5.70±0.06 | **4.55±0.04** |
| | Efficientnet-b1 | 6.78±0.07 | 6.12±0.15 | 3.99±0.07 | 6.54±0.06 | 4.87±0.05 | **4.05±0.03** |
| | BiT-M-R50 | 6.93±0.27 | 6.99±0.25 | 3.86±0.06 | 6.32±0.16 | 6.50±0.16 | **4.30±0.04** |
| | ViT-Base | 4.77±0.16 | 4.34±0.12 | 3.76±0.07 | 4.09±0.06 | 4.01±0.05 | **3.86±0.04** |
| WILDS-RxRx1 | ResNet-50 | 26.22±0.38 | 9.83±0.57 | 2.85±0.17 | 26.22±0.38 | 13.78±0.43 | **5.25±0.11** |
| | ResNext-50 | 25.30±0.76 | 9.39±0.58 | 3.13±0.19 | 20.71±0.30 | 11.80±0.37 | **5.07±0.09** |
| | DenseNet-121 | 32.37±0.91 | 8.91±0.60 | 2.94±0.18 | 24.49±0.35 | 13.08±0.41 | **5.38±0.13** |
| GLDv2 | ResNet-50 | 12.56±0.08 | 11.61±0.09 | 9.90±0.06 | 11.36±0.15 | 10.75±0.14 | **9.76±0.12** |
| | BiT-M-R50 | 14.86±0.12 | 11.31±0.07 | 9.78±0.06 | 13.91±0.21 | 9.83±0.11 | **9.16±0.10** |
| | ViT-Small | 12.44±0.11 | 11.12±0.07 | 9.75±0.05 | 11.00±0.18 | 9.65±0.11 | **9.01±0.10** |

benchmark image classification dataset, WILDS-RxRx1 [Koh et al., 2021]—an image of cells (by fluorescent microscopy) dataset in the domain generalization benchmark, and GLDv2 [Weyand et al., 2020]—a landmark recognition dataset in federated learning. Additional experimental results and implementation details can be found in Appendix C. Our code is available at `https://github.com/yaodongyu/MDTS`.

**Datasets.** We evaluate different calibration methods on three datasets, ImageNet-C, WILDS-RxRx1, and GLDv2. ImageNet-C contains 15 types of common corruptions where each corruption includes five severity levels. Each corruption with one severity is one domain, and there are 76 domains in total (including the standard ImageNet validation dataset). We partition the 76 domains into disjoint in-distribution domains and out-of-distribution by severity level or corruption type. WILDS-RxRx1 is a domain generalization dataset, and we treat each experimental domain as one domain. We adopt the default val/test split in Koh et al. [2021]: use the four validation domains as in-distribution domains and the 14 test domains as the out-of-distribution domains. We also provide experimental results of other random splits in Appendix C. For GLDv2, each client corresponds to one domain, and there are 823 domains in total. We randomly select 500 domains for training the model, and then use the remaining 323 domains for evaluation denoted by validation domains. We further screen the validation domains by removing the domains with less than 300 data points. There are 44 domains after screening, and we use 30 domains as in-distribution domains and the remaining 14 domains as out-of-distribution domains. For all datasets, we randomly sample half of the data from in-distribution domains for calibrating models and use the remaining samples for InD ECE evaluation. We use all the samples from OOD domains for ECE evaluation.

**Models and training setup.** We consider multiple network architectures for evaluation, including ResNet-50 [He et al., 2016], ResNext-50 [Xie et al., 2017], DenseNet-121 [Huang et al., 2017], BiT-M-50 [Kolesnikov et al., 2020], Efficientnet-b1 [Tan and Le, 2019], ViT-Small, and ViT-Base [Dosovitskiy et al., 2020]. To evaluate on ImageNet-C, we directly evaluate models that are pre-trained on ImageNet [Deng et al., 2009]. For WILDS-RxRx1 and GLDv2, we use the ImageNet pre-trained models as initialization and apply SGD optimizer to training the models on training datasets.

**Evaluation metrics.** We use the Expected Calibration Error (ECE) as the main evaluation metric. We set the bin size as 100 for ImageNet-C, and set bin size as 20 for WILDS-RxRx1 and GLDv2. We evaluate ECE on both InD domains and OOD domains. Specifically, we evaluate the ECE of each InD/OOD domain. Meanwhile, we also evaluate the ECE of the pooled InD/OOD domains, i.e., the ECE evaluated on all samples from InD/OOD domains. We use unseen samples from the InD domain to measure the per-domain ECE. We also measure the averaged per-domain ECE results (i.e., per-domain ECE averaged across domains).

## 4.1 Main results

We summarize the ECE results of different methods on three datasets in Table 1 and Figure 3. We use TS to denote temperature scaling [Guo et al., 2017], and use MSP to denote applying the maximum softmax probability [Hendrycks and Gimpel, 2016] of the model output (i.e., without calibration).

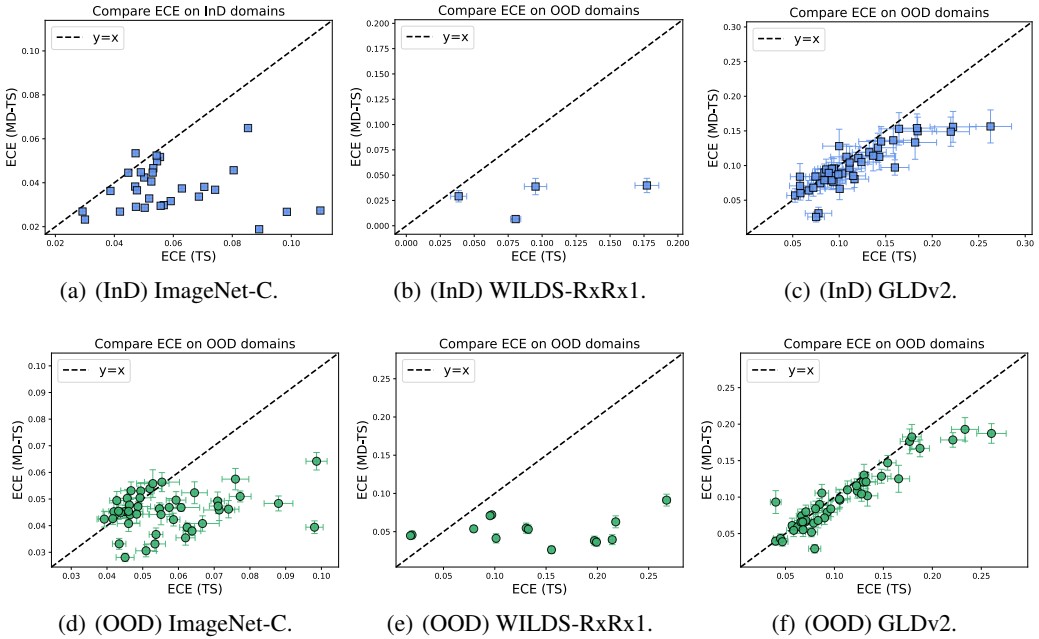

| (a) (InD) ImageNet-C. | (b) (InD) WILDS-RxRx1. | (c) (InD) GLDv2. |
| (d) (OOD) ImageNet-C. | (e) (OOD) WILDS-RxRx1. | (f) (OOD) GLDv2. |

Figure 3: Per-domain ECE of MD-TS and TS on both in-distribution domains and out-of-distribution domains. Each plot is shown with ECE of TS ($X$-axis) and ECE of MD-TS ($Y$-axis). Top: per-domain ECE evaluated on InD domains. Bottom: per-domain ECE evaluated on OOD domains. Lower ECE is better.

In Table 1, we use the ImageNet validation dataset and ImageNet-C datasets with severity level $s \in \{1, 5\}$ as the InD domains and use the remaining datasets as OOD domains. We present the averaged per-domain ECE results in Table 1, and visualize the ECE of each domain in Figure 3. As shown in Table 1 and Figure 3(a)-3(c), we find that our proposed approach achieves much better InD per-domain calibration compared with baselines. Also, TS does not significantly improve over MSP on ImageNet-C InD domains in Table 1, but our proposed method largely improve the ECE compared with MSP and TS. For instance, the ECE results of MSP and TS on Efficientnet-b1 are 6.93 and 6.99, and our method achieves 3.84. Intuitively, when there are a diverse set of domains in the calibration dataset, a single temperature cannot provide well-calibrated confidences. In contrast, our proposed method is able to produce much better InD confidence estimates by leveraging the multi-domain structure of the data.

Next we study the performance of different methods on out-of-distribution domains. From Table 1, we find that MD-TS achieves the best performance on OOD domains arcoss all the settings. On ImageNet-C with BiT-M-R50, MD-TS improves the ECE from 6.54 (MSP) to 4.05, while the performance of TS is similar to MSP. Moreover, MD-TS significantly outperforms MSP and TS on WILDS-RxRx1, where MD-TS improves over TS by around 5.00 measured in ECE. Figure 3(d)-3(f) display the per-domain ECE performance on out-of-distribution domains. MD-TS improves over TS on more than half of the domains in all three datasets. For the remaining domains, MD-TS performs slightly worse than TS. Furthermore, on those domains that TS performs poorly (ECE > 8), MD-TS largely improves over TS by large margins. Further comparisons in Appendix C.6 show that these improvements continue to hold when relative to two other calibration techniques: MC dropout [Gal and Ghahramani, 2016] and deep ensembles [Lakshminarayanan et al., 2017].

## 4.2 Predicting generalization

Suppose a model can produce calibrated confidences on unseen samples, in which case we could leverage the calibrated confidence to predict the model performance. Specifically, based on the definition of ECE in Eq. (1), when the model is well-calibrated, the average of the calibrated confidence is close to the average accuracy, i.e., $\mathsf{Conf}(\mathcal{D}) \approx \mathsf{Acc}(\mathcal{D})$.[2] Meanwhile, predicting model

---

[2]$\mathsf{Conf}(\mathcal{D})$ denotes the average (calibrated) confidence on dataset $\mathcal{D}$, and $\mathsf{Acc}(\mathcal{D})$ denotes the average accuracy on dataset $\mathcal{D}$.

Table 2: Model performance prediction comparison results of different methods on three datasets. Lower MAE indicates better performance.

| Datasets | Architectures | InD-domains MAE | | | OOD-domains MAE | | |
|---|---|---|---|---|---|---|---|
| | | MSP | TS | MD-TS | MSP | TS | MD-TS |
| ImageNet-C | ResNet-50 | 5.88 | 4.74 | 1.28 | 5.15 | 3.96 | **1.70** |
| | BiT-M-R50 | 6.08 | 6.16 | 1.33 | 4.97 | 5.23 | **1.66** |
| WILDS-RxRx1 | ResNet-50 | 33.65 | 9.61 | 1.61 | 26.20 | 13.66 | **4.76** |
| | ResNext-50 | 25.32 | 8.55 | 1.39 | 20.72 | 12.88 | **4.78** |
| GLDv2 | ResNet-50 | 9.60 | 9.17 | 7.11 | 9.72 | 9.40 | **8.08** |
| | BiT-M-R50 | 12.67 | 7.18 | 4.64 | 12.30 | 7.34 | **6.37** |

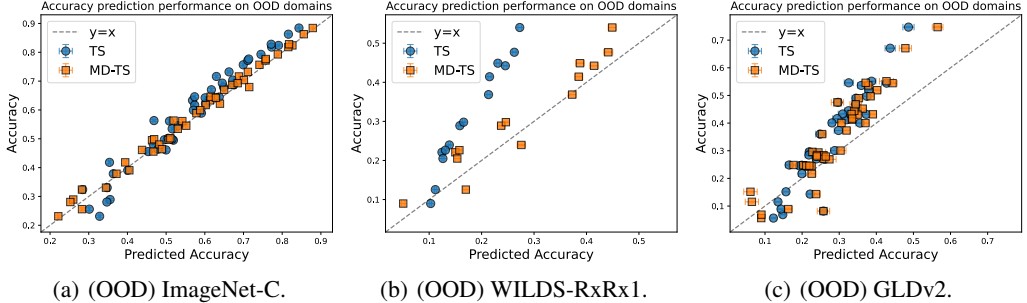

(a) (OOD) ImageNet-C.  (b) (OOD) WILDS-RxRx1.  (c) (OOD) GLDv2.

Figure 4: Predicting accuracy performance of MD-TS and TS on both out-of-distribution domains. Each plot is shown with predicted accuracy ($X$-axis) and accuracy ($Y$-axis). Each points corresponds to one domain. The network architecture is ResNet-50 for three datasets. Point closer to the $Y = X$ dashed line means better prediction performance.

performance accurately is an essential ingredient in developing reliable machine learning systems, especially under distributional shifts [Guillory et al., 2021]. As shown in Table 1, we find that our proposed method produces well-calibrated confidence values on both InD and OOD domains. We now measure its performance on predicting model performance and compare with existing methods. We measure the performance using mean absolute error (MAE), $\mathrm{MAE} = (1/K) \cdot \sum_{k=1}^{K} |\mathsf{Conf}(\mathcal{D}_k) - \mathsf{Acc}(\mathcal{D}_k)|$ where $S_k$ is the dataset from the $k$-th domain.

We show the predicting model accuracy results in Table 2. MD-TS significantly improves over existing methods on predicting model performance across all three datasets. For example, on ImageNet-C, calibrated confidence of MD-TS produces fairly accurate predictions on both InD and OOD domains (less than 2% measured in MAE), which largely outperforms MSP and TS. In Figure 4, we compared the prediction performance of TS and MD-TS on every OOD domain. We find that MD-TS achieves better prediction performance compared to TS on most of the domains. Refer to Appendix C.1 for more results in which other architectures are tested.

### 4.3 MD-TS ablations

To learn a calibration model that performs well per-domain, we apply linear regression on feature representations $\Phi(x_k)$ such that $\langle \Phi(x_k), \theta \rangle \approx \hat{T}_k$, where $x_k$ is from domain $k$ and $\hat{T}_k$ is the temperature parameter for domain $k$. We investigate other methods for learning the map from feature representations to temperatures in a regression framework. Specifically, beside the ordinary least squares (OLS) used in Algorithm 1, we consider ridge regression (Ridge), robust regression with Huber loss (Huber), kernel ridge regression (KRR), and $K$-nearest neighbors regression (KNN). The implementations are mainly based on `scikit-learn` [Pedregosa et al., 2011]. We use grid search (on InD domains) to select hyperparameters for Ridge, Huber, KRR, and KNN.

We summarize the comparative results for different regression algorithms in Table 3. Compared to OLS, other regression algorithms do not achieve significant improvement. Specifically, KRR achieves slightly better performance on OOD domains, while other algorithms have similar performance compared to OLS. Moreover, there are no hyperparameter in OLS, which makes it more practical in

Table 3: Per-domain ECE (%) results of MD-TS ablations on WILDS-RxRx1. We evaluate the per-domain ECE on InD and OOD domains, and report the mean and standard error of per-domain ECE. Lower ECE means better performance.

| Architectures | InD-domains | | | | | OOD-domains | | | | |
|---|---|---|---|---|---|---|---|---|---|---|
| | OLS | Ridge | Huber | KRR | KNN | OLS | Ridge | Huber | KRR | KNN |
| ResNet-50 | 2.85 | 2.88 | 2.90 | 2.85 | 3.00 | 5.25 | 5.26 | 5.29 | 4.99 | 5.44 |
| ResNext-50 | 3.13 | 3.14 | 3.11 | 3.07 | 3.03 | 5.07 | 5.06 | 5.02 | 4.94 | 5.36 |
| DenseNet-121 | 2.94 | 3.03 | 2.92 | 2.90 | 3.04 | 5.38 | 5.42 | 5.36 | 5.20 | 5.47 |

real-world problems. Meanwhile, the results suggest that our proposed MD-TS is stable to the choice of specific regression algorithms.

## 5 Theoretical analysis

In this section, we provide theoretical analysis to support our understanding of our proposed algorithm in the presence of distribution shifts. We use $h_k^\star(\cdot) = h(\cdot; f, \beta_k^\star) : \mathcal{X} \to [0, 1]$ to denote the best calibration map for the base model $f$ on the $k$-th domain; this map *minimizes* the expected calibration error (ECE) $\mathbb{E}[|p - \mathbb{P}(\hat{y} = y | \hat{\pi} = p)|]$ over distribution $\mathcal{P}_k$. We also call $h_k^\star$ a hypothesis in the hypothesis class $\mathcal{H}$. Next, given the fixed base model $f$, we aim to learn $\hat{h}(\cdot) = h(\cdot; f, \hat{\beta})$ such that $\varepsilon(\hat{h}, \mathcal{P}_{k,X}) = \mathbb{E}_{X \sim \mathcal{P}_{k,X}}[|h_k^\star(X) - \hat{h}(X)|]$ is small for *every* domain $k$, where $\varepsilon_k(\hat{h})$ denotes the risk of $\hat{h}$ w.r.t. the the best calibration map $h_k^\star$ under domain $\mathcal{P}_k$. In addition, we are interested in generalizing to new domains: suppose there is an unseen OOD domain $\widetilde{\mathcal{P}}$ and its marginal feature distribution is different from existing domains, i.e., $\widetilde{\mathcal{P}}_X \neq \mathcal{P}_{k,X}$ for $k \in [K]$.

Our goal is to understand the conditions under which $\hat{h}$ can have similar calibration on OOD domains as the InD domains. For example, if the OOD domain is similar to the mixture distribution of InD domains, we would expect $\hat{h}$ performs similarly on InD and OOD domains. Previous work [Krueger et al., 2021] consider a similar multiple training domains setting in the context of out-of-distribution generalization, where they show that minimizing the differences in training risks (w.r.t. different domains) can lead to good OOD performance. To quantify the distance between two distributions, we first introduce the $\mathcal{H}$-divergence [Ben-David et al., 2010] to measure the distance between two distributions:

**Definition 5.1** ($\mathcal{H}$-divergence). *Given an input space $\mathcal{X}$ and two probability distributions $\mathcal{P}_X$ and $\mathcal{P}_X'$ on $\mathcal{X}$, let $\mathcal{H}$ be a hypothesis class on $\mathcal{X}$, and denote by $\mathcal{A}$ the collection of subsets of $\mathcal{X}$ which are the support of hypothesis $h \in \mathcal{H}$, i.e., $\mathcal{A}_\mathcal{H} = \{h^{-1}(1) \,|\, h \in \mathcal{H}\}$. The distance between $\mathcal{P}_X$ and $\mathcal{P}_X'$ is defined as*

$$d_\mathcal{H}(\mathcal{P}_X, \mathcal{P}_X') = \sup_{A \in \mathcal{A}_\mathcal{H}} \left| \mathrm{Pr}_{\mathcal{P}_X}(A) - \mathrm{Pr}_{\mathcal{P}_X'}(A) \right|.$$

The $\mathcal{H}$-divergence reduces to the standard total variation (TV) distance when $\mathcal{H}$ contains all measurable functions on $\mathcal{X}$, which implies that the $\mathcal{H}$-divergence is upper bounded by the TV-distance, i.e., $d_\mathcal{H}(\mathcal{P}_X, \mathcal{P}_X') \leq d_{\mathsf{TV}}(\mathcal{P}_X, \mathcal{P}_X')$. On the other hand, when the hypothesis class $\mathcal{H}$ has a finite VC dimension or pseudo-dimension, the $\mathcal{H}$-divergence can be estimated using finite samples from $\mathcal{P}_X$ and $\mathcal{P}_X'$ [Ben-David et al., 2010]. Next, we define the mixture distribution of the $K$ in-distribution domains $\mathcal{P}_{K,X}^\alpha$ on input space $\mathcal{X}$ as follows:

$$\mathcal{P}_{K,X}^\alpha = \sum_{k=1}^K \alpha_k \mathcal{P}_{k,X}, \quad \text{where} \quad \sum_{k=1}^K \alpha_k = 1 \text{ and } \alpha_k \geq 0.$$

Given multiple domains $\{\mathcal{P}_1, \ldots, \mathcal{P}_K\}$, we can optimize the combination parameters $\alpha$ such that $\mathcal{P}_{K,X}^\alpha$ minimizes the $\mathcal{H}$-divergence between $\mathcal{P}_{K,X}^\alpha$ and $\widetilde{\mathcal{P}}_X$. More specifically, we define $\hat{\alpha}$ as

$$\hat{\alpha} = \operatorname*{argmin}_{\alpha \in \Delta} \left\{ \frac{1}{2} d_{\bar{\mathcal{H}}}(\mathcal{P}_{K,X}^\alpha, \widetilde{\mathcal{P}}_X) + \lambda(\mathcal{P}_{K,X}^\alpha, \widetilde{\mathcal{P}}_X) \right\}, \quad \lambda(\mathcal{P}_{K,X}^\alpha, \widetilde{\mathcal{P}}_X) = \varepsilon(h^\star, \mathcal{P}_{K,X}^\alpha) + \varepsilon(h^\star, \widetilde{\mathcal{P}}_X), \quad (3)$$

where $h^\star := \operatorname{argmin}_{h \in \mathcal{H}} \{\varepsilon(h, \mathcal{P}^\alpha_{K,X}) + \varepsilon(h, \widetilde{\mathcal{P}}_X)\}$ and $\bar{\mathcal{H}}$ is defined as $\bar{\mathcal{H}} := \{\operatorname{sign}(|h(x) - h'(x)| - t) \mid h, h' \in \mathcal{H}, 0 \le t \le 1\}$. We now give an upper bound on the risk on the unseen OOD domain. This result follows very closely those of Blitzer et al. [2007]; Zhao et al. [2018], instantiated in our calibration setup. Details can be found in Appendix D.

**Theorem 5.2.** *Let $\mathcal{H}$ be a hypothesis class that contains functions $h : \mathcal{X} \to [0, 1]$ with pseudo-dimension $\operatorname{Pdim}(\mathcal{H}) = d$. Let $\{D_{k,X}\}_{k=1}^K$ denote the empirical distributions generated from $\{\mathcal{P}_{k,X}\}_{k=1}^K$, where $\mathcal{D}_{k,X}$ contains $n$ i.i.d. samples from the marginal feature distribution $\mathcal{P}_{k,X}$ of domain $k$. Then for $\delta \in (0, 1)$, with probability at least $1 - \delta$, we have*

$$\varepsilon(\hat{h}, \widetilde{\mathcal{P}}_X) \le \sum_{k=1}^K \hat{\alpha}_k \cdot \hat{\varepsilon}(\hat{h}, \mathcal{D}_{k,X}) + \frac{1}{2} \operatorname{d}_{\bar{\mathcal{H}}}(\mathcal{P}^{\hat{\alpha}}_{K,X}, \widetilde{\mathcal{P}}_X) + \lambda(\mathcal{P}^{\hat{\alpha}}_{K,X}, \widetilde{\mathcal{P}}_X) + \widetilde{O}\left(\frac{\operatorname{Pdim}(\mathcal{H})}{\sqrt{nK}}\right), \quad (4)$$

*where $\hat{\alpha}$ and $\lambda(\mathcal{P}^{\hat{\alpha}}_{K,X}, \widetilde{\mathcal{P}}_X)$ are defined in Eq. (3), $\widetilde{\mathcal{P}}_X$ denotes the marginal distribution of the OOD domain, $\operatorname{Pdim}(\mathcal{H})$ is the pseudo-dimension of the hypothesis class $\mathcal{H}$, and $\hat{\varepsilon}(\hat{h}, \mathcal{D}_{k,X})$ is the empirical risk of the hypothesis $\hat{h}$ on $\mathcal{D}_{k,X}$.*

**Remark 5.3.** *As shown in Theorem 5.2, even if the OOD domain is very different from the in-distribution domains, the Eq. (4) still implies that we could decrease the risk upper bound on the OOD domain if we perform multi-domain calibration. More specifically, if we could achieve good calibration performance on each individual domain by using multi-domain calibration (which that the first term in the RHS of Eq. (4) is small), then the term $\frac{1}{2} \operatorname{d}_{\bar{\mathcal{H}}}(\mathcal{P}^{\hat{\alpha}}_{K,X}, \widetilde{\mathcal{P}}_X) + \lambda(\mathcal{P}^{\hat{\alpha}}_{K,X}, \widetilde{\mathcal{P}}_X)$ is always smaller or equal to $\frac{1}{2} \operatorname{d}_{\bar{\mathcal{H}}}(\mathcal{P}', \widetilde{\mathcal{P}}_X) + \lambda(\mathcal{P}', \widetilde{\mathcal{P}}_X)$, where $\mathcal{P}'$ is the pooled distribution or any individual domain distribution.*

**Remark 5.4.** *As suggested by Eq. (4) of Theorem 5.2, larger risks on in-distribution domains will lead to a larger upper bound for the risk evaluated on the OOD domain. On the other hand, as shown in Figure 1, a universal temperature is not sufficient to achieve good calibration performance on each individual in-distribution domain. Therefore, even in the mixture of in-distribution domain setting, a universal temperature is suboptimal and applying multi-domain temperature scaling could be better than using a universal temperature.*

This result means that if we can learn a hypothesis $\hat{h}$ that achieves small empirical risk $\hat{\varepsilon}(\hat{h}, \mathcal{D}_{k,X})$ on every domain, then $\hat{h}$ is able to achieve good performance on the OOD domain if distribution of the OOD domain is similar to the mixture distribution of InD domains measured by $\mathcal{H}$-divergence. In this case, if the learned calibration map $\hat{h}$ is well-calibrated on every domain $\mathcal{P}_k$, then $\hat{h}$ is likely to provide calibrated confidence for the OOD domain $\widetilde{\mathcal{P}}$. Recall from Section 4, we proposed an algorithm that performs well across InD domains. The upper bound in Eq. (4) provides insight into understanding why this algorithm is effective.

## 6 Discussion

We have developed an algorithm for robust calibration that exploits multi-domain structure in datasets. Experiments on real-world domains indicate that multi-domain calibration is an effective way to improve the robustness of calibration under distribution shifts. Our proposed algorithm still needs validation domains to achieve strong calibration performance on OOD domains, one interesting direction for future work would be to extend our algorithm to a scenario where no domain information is available. We hope the multi-domain calibration perspective in this paper can motivate further work to close the gap between in-distribution and out-of-distribution calibration.

## Acknowledgments and Disclosure of Funding

We would like to thank the anonymous reviewers for their constructive suggestions and comments. Yaodong Yu acknowledges support from the joint Simons Foundation-NSF DMS grant #2031899. Stephen Bates was supported by the Foundations of Data Science Institute and the Simons Institute. Yi Ma acknowledges support from ONR grants N00014-20-1-2002 and N00014-22-1-2102 and the joint Simons Foundation-NSF DMS grant #2031899. Michael Jordan acknowledges support from the ONR Mathematical Data Science program.

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
