# Appendix

## A    Additional Details

**Pseudocode for MD-TS.**    We first provide additional details on our proposed algorithm, MD-TS (**M**ult-**D**omain **T**emperature **S**caling), in Algorithm 1.

---
**Algorithm 1** MD-TS

---
**Input:** Data from $k$-th domain $\mathcal{D}_k = \{(x_{i,k}, y_{i,k})\}_{i=1}^{n_k}$, $k \in [K]$, base model $f$, feature embedding map of base model $\Psi$, and the test sample $\widetilde{x}$.

1: **for** $k = 1, \ldots, K$ **do**
2:    $\hat{T}_k = \texttt{TS-Alg}(\mathcal{D}_k, f)$
3: **end for**
4: Learn the linear model, $\hat{\theta} = \underset{\theta}{\operatorname{argmin}} \sum_{k=1}^{K} \sum_{i=1}^{n_k} \left( \langle \Psi(x_{i,k}), \theta \rangle - \hat{T}_k \right)^2$.
5: Predict temperature $\widetilde{T}$ the test sample $\widetilde{x}$ using the learned linear model, $\widetilde{T} = \langle \Psi(x_{i,k}), \hat{\theta} \rangle$
6: Compute the confidence estimate for sample $\widetilde{x}$ as $\widetilde{\pi} = \max_j \left[ \operatorname{Softmax}\left( f(\widetilde{x}) / \widetilde{T} \right) \right]_j$.

**Output:** Confidence estimate $\widetilde{\pi}$ for the test sample $\widetilde{x}$.

---

**Experimental details (checklist).**    We provide additional details about the training and compute. Details about data splits and hyperparameters for training can be found in Section 4 and Appendix C. We use NVIDIA 2080Ti and A100 GPUs, and our experiments required around 100 hours of GPU time.

## B    Societal Impact

In this paper, we aim to improve the trustworthiness of machine learning systems by first, explicitly accounting for uncertainty, and second, do this in a way that is robust to distribution shifts. As uncertainty quantification is an increasingly important component of real-world machine learning systems, including health care and autonomous driving, we believe our work could potentially benefit a wide range of societal activities. Moreover, our method explicitly takes into account subgroups of the data, trying to achieve good performance across all data. It is known that this is an important aspect of performance when deploying models in high-consequence settings [Barocas et al., 2019]. We hope our work could offer a new perspective on uncertainty quantification under distributional shifts. We do not anticipate the negative social impact of this work.

# C Additional Experimental Results

In this section, we provide additional implementation details and experimental results.

**Details about dataset and pre-trained model.** For ImageNet, we consider a 200 classes subset of ImageNet. Details can be found in Hendrycks et al. [2021]. The Efficientnet-b1 is trained by AdvProp and AutoAugment data augmentation [Xie et al., 2020].

## C.1 Additional experimental results of predicting model accuracy

We provide additional results (other network architectures) of the prediction performance of TS and MD-TS on every OOD domain in Figure 5.

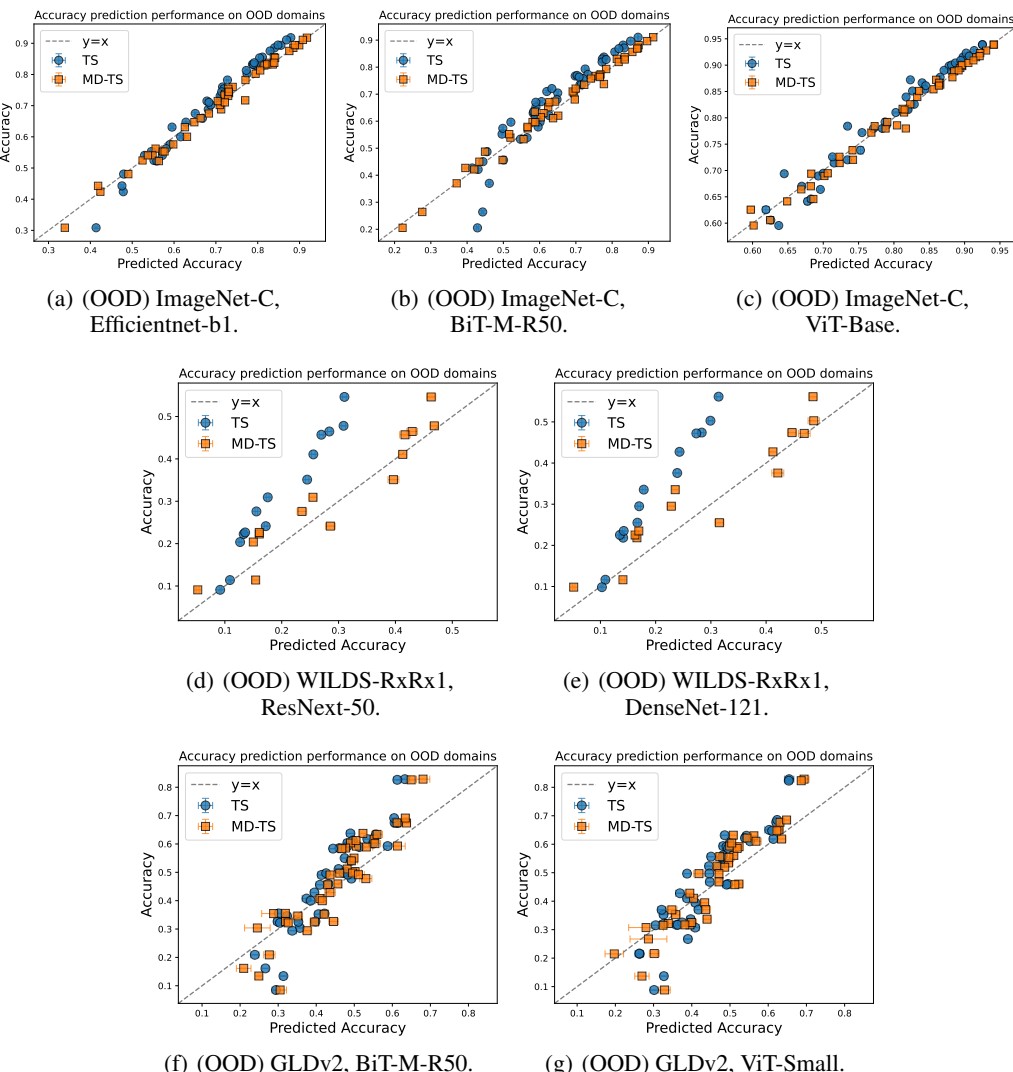

Figure 5: (*Evaluated on more network architectures*) Predicting accuracy performance of MD-TS and TS on both out-of-distribution domains. Each plot is shown with predicted accuracy ($X$-axis) and accuracy ($Y$-axis). Each point corresponds to one domain. The network architecture is ResNet-50 for three datasets. Point closer to the $Y = X$ dashed line means better prediction performance.

## C.2 Additional experimental results of WILDS-RxRx1

We provide additional MDECE results of TS and MD-TS on WILDS-RxRx1 (evaluated on other InD/OOD splits) in Figure 6. For every InD/OOD split, we randomly sample 4 domains as the InD domains and set the remaining domains as OOD domains.

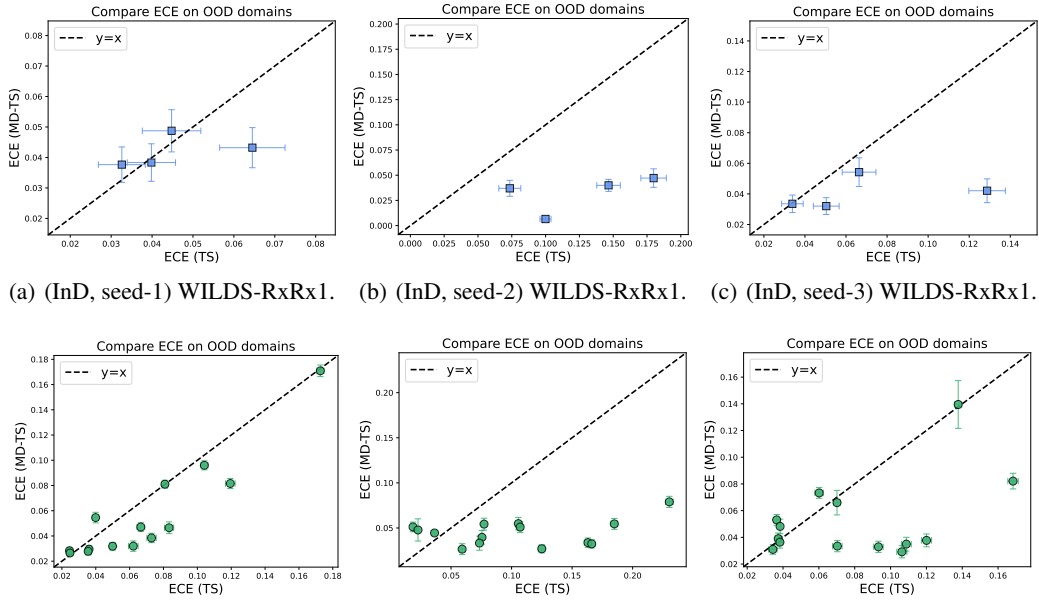

(a) (InD, seed-1) WILDS-RxRx1.  (b) (InD, seed-2) WILDS-RxRx1.  (c) (InD, seed-3) WILDS-RxRx1.

(d) (OOD, seed-1) WILDS-RxRx1. (e) (OOD, seed-2) WILDS-RxRx1. (f) (OOD, seed-3) WILDS-RxRx1.

Figure 6: (*Evaluated on WILDS-RxRx1 with other InD/OOD splits.*) Per-domain ECE of MD-TS and TS on both in-distribution domains and out-of-distribution domains. Each plot is shown with ECE of TS ($X$-axis) and ECE of MD-TS ($Y$-axis). Top: per-domain ECE evaluated on InD domains. Bottom: per-domain ECE evaluated on OOD domains. Lower ECE is better.

## C.3 Experimental results of overall ECE

We provide the overall ECE results, evaluated on the *pooled* InD/OOD data, of different methods on three datasets in Table 4.

Table 4: ECE (%) comparison on three datasets. We evaluate the ECE on *pooled* InD and OOD domains. We report the mean and standard error of ECE on one dataset. Lower ECE means better performance.

| Datasets | Architectures | InD-domains | | | OOD-domains | | |
|---|---|---|---|---|---|---|---|
| | | MSP | TS | MD-TS | MSP | TS | MD-TS |
| ImageNet-C | ResNet-50 | 5.40±0.03 | 1.09±0.03 | 1.06±0.02 | 4.86±0.04 | 2.27±0.06 | **1.89±0.03** |
| | Efficientnet-b1 | 2.88±0.03 | 1.23±0.04 | 1.52±0.06 | 5.21±0.03 | 1.56±0.05 | **1.43±0.03** |
| | BiT-M-R50 | 0.90±0.02 | 0.89±0.02 | 1.18±0.05 | 2.04±0.04 | 2.53±0.07 | **1.52±0.04** |
| | ViT-Base | 2.31±0.03 | 1.40±0.02 | 1.90±0.04 | 2.04±0.02 | **1.91±0.02** | 2.09±0.03 |
| WILDS-RxRx1 | ResNet-50 | 33.57±0.07 | 5.77±0.11 | 2.01±0.07 | 26.18±0.00 | 13.51±0.07 | **3.67±0.08** |
| | ResNext-50 | 25.26±0.08 | 5.90±0.11 | 2.27±0.07 | 20.71±0.00 | 11.62±0.07 | **2.77±0.08** |
| | DenseNet-121 | 32.27±0.09 | 5.05±0.13 | 1.87±0.09 | 24.48±0.00 | 12.83±0.08 | **2.95±0.09** |
| GLDv2 | ResNet-50 | 9.02±0.08 | 6.87±0.07 | 5.93±0.06 | 8.90±0.16 | 7.10±0.25 | **6.14±0.21** |
| | BiT-M-R50 | 12.20±0.10 | 3.65±0.05 | 2.99±0.05 | 12.25±0.21 | 4.18±0.20 | **3.53±0.15** |
| | ViT-Small | 7.79±0.09 | 3.09±0.05 | 2.61±0.04 | 7.79±0.20 | 3.89±0.16 | **3.41±0.12** |

## C.4 Experimental results of other calibration methods

We provide the MDECE results of other calibration methods, including histogram binning (Hist-Bin) [Zadrozny and Elkan, 2001], isotonic regression (Isotonic) [Zadrozny and Elkan, 2002], and Bayesian Binning into Quantiles (BBQ) [Naeini et al., 2015], on three datasets in Table 5. By comparing the results in Table 5 and Table 1, we find that our algorithm largely outperforms these methods on three datasets.

Table 5: Per-domain ECE (%) comparison of histogram binning (HistBin), isotonic regression (Isotonic), and Bayesian Binning into Quantiles (BBQ) on three datasets. We evaluate the per-domain ECE on InD and OOD domains. We report the mean and standard error of per-domain ECE on one dataset. Lower ECE means better performance.

| Datasets | Architectures | InD-domains | | | OOD-domains | | |
|---|---|---|---|---|---|---|---|
| | | HistBin | Isotonic | BBQ | HistBin | Isotonic | BBQ |
| ImageNet-C | ResNet-50 | 9.50±0.25 | 5.17±0.12 | 13.87±0.03 | 9.17±0.17 | 5.23±0.07 | 11.63±0.16 |
| | Efficientnet-b1 | 7.10±0.22 | 5.30±0.06 | 13.39±0.15 | 5.56±0.10 | 4.94±0.04 | 11.88±0.13 |
| | BiT-M-R50 | 7.31±0.27 | 6.44±0.16 | 12.95±0.21 | 6.05±0.16 | 6.79±0.09 | 12.00±0.15 |
| | ViT-Base | 6.85±0.26 | 4.30±0.03 | 12.26±0.12 | 5.35±0.10 | 3.97±0.02 | 10.74±0.06 |
| WILDS-RxRx1 | ResNet-50 | 11.64±0.23 | 11.67±0.42 | 6.56±0.44 | 11.89±0.22 | 8.99±0.28 | 11.31±0.36 |
| | ResNext-50 | 10.10±0.17 | 11.39±0.30 | 6.70±0.59 | 11.25±0.16 | 9.54±0.23 | 11.79±0.41 |
| | DenseNet-121 | 11.95±0.24 | 11.97±0.13 | 6.87±0.54 | 12.04±0.19 | 8.72±0.24 | 11.93±0.06 |
| GLDv2 | ResNet-50 | 14.77±0.16 | 11.43±0.07 | 15.32±0.20 | 10.72±0.08 | 16.96±0.17 | 10.04±0.10 |
| | BiT-M-R50 | 15.24±0.12 | 10.93±0.07 | 16.10±0.15 | 12.93±0.11 | 20.86±0.21 | 12.15±0.14 |
| | ViT-Small | 15.47±0.13 | 10.86±0.08 | 16.20±0.15 | 12.12±0.10 | 19.89±0.20 | 11.61±0.14 |

## C.5 Additional experimental results of ImageNet-C

We consider a different InD/OOD partition from the ones in Section 4. Specifically, we use the ImageNet validation dataset and ImageNet-C datasets with severity level $s \in \{1, 2, 3, 4\}$ as the InD domains and use the remaining datasets as OOD domains. The results are summarized in Table 6. This is a more challenging setting since it requires calibration methods to extrapolate to a higher severity level. As shown in Table 6, our method still achieves the best performance in all settings, except for Efficientnet-b1 (OOD-domains). This is possibly because the Efficientnet-b1 model is pre-trained with AdvProp and AutoAugment data, it achieves better performance on corruptions with high severity levels than standard pre-trained models.

Table 6: Per-domain ECE (%) comparison on ImageNet-C datasets (with InD/OOD split mentioned in Section C.5). We evaluate the per-domain ECE on InD and OOD domains. We report the mean and standard error of per-domain ECE on one dataset. Lower ECE means better performance.

| Datasets | Architectures | InD-domains | | | OOD-domains | | |
|---|---|---|---|---|---|---|---|
| | | MSP | TS | MD-TS | MSP | TS | MD-TS |
| ImageNet-C | ResNet-50 | 5.99±0.14 | 5.11±0.07 | 4.17±0.03 | 11.45±0.36 | 7.97±0.25 | **5.89±0.20** |
| | Efficientnet-b1 | 6.71±0.05 | 4.41±0.09 | 3.97±0.05 | **6.37±0.16** | 10.45±0.37 | 8.31±0.25 |
| | BiT-M-R50 | 5.96±0.12 | 5.21±0.16 | 3.98±0.04 | 9.05±0.50 | 11.01±0.61 | **7.71±0.25** |
| | ViT-Base | 3.72±0.05 | 3.78±0.06 | 3.62±0.05 | 7.02±0.23 | 7.33±0.25 | **6.16±0.13** |

## C.6 Compare to additional existing methods

We compare our proposed method with *Ensembles* [Lakshminarayanan et al., 2017] and *Dropout* [Gal and Ghahramani, 2016] , and the results are summarized in Table 7,

Table 7: Per-domain ECE (%) comparison on two datasets. We evaluate the per-domain ECE on InD and OOD domains. We report the mean and standard error of per-domain ECE on one dataset. Lower ECE means better performance. We use *Ensembles* to denote the deep ensembles method proposed in Lakshminarayanan et al. [2017], and *Dropout* to denote the Monte-Carlo Dropout method proposed in Gal and Ghahramani [2016].

| Datasets | Architectures | InD-domains | | | OOD-domains | | |
|---|---|---|---|---|---|---|---|
| | | Ensembles | Dropout | MD-TS | Ensembles | Dropout | MD-TS |
| WILDS-RxRx1 | ResNet-50 | 14.73±0.67 | 18.59±0.58 | 2.85±0.17 | 8.80±0.25 | 11.72±0.44 | **5.25±0.11** |
| GLDv2 | ResNet-50 | 10.85±0.08 | 11.73±0.13 | 9.90±0.06 | 9.98±0.12 | 10.43±0.20 | **9.76±0.12** |

We also study the performance of applying a 2-layer MLP, which has been used in Kumar and Sarawagi [2019], instead of the linear model in our proposed approach for predicting the temperature. We do not find that using 2-layer MLP lead to better performance compared to the linear model approach. For example, on WILDS-RxRx1 with ResNet-50, the per-domain ECE of the 2-layer MLP is 5.62% whereas per-domain ECE of the linear model is 5.25%.

## C.7 Additional experimental results on the NLP dataset

We study the performance of our proposed method on WILDS-Amazon, and we present the results in Table 8 and Figure 7.

Table 8: Per-domain ECE (%) comparison on WILDS-Amazon [Koh et al., 2021]. We evaluate the per-domain ECE on InD and OOD domains. We report the mean and standard error of per-domain ECE on one dataset. Lower ECE means better performance.

| Datasets | Architectures | InD-domains | | | OOD-domains | | |
|---|---|---|---|---|---|---|---|
| | | MSP | TS | MD-TS | MSP | TS | MD-TS |
| WILDS-Amazon | DistilBERT | 31.88±0.22 | 10.16±0.22 | 9.83±0.20 | 29.89±0.18 | 7.77±0.16 | **7.59±0.14** |

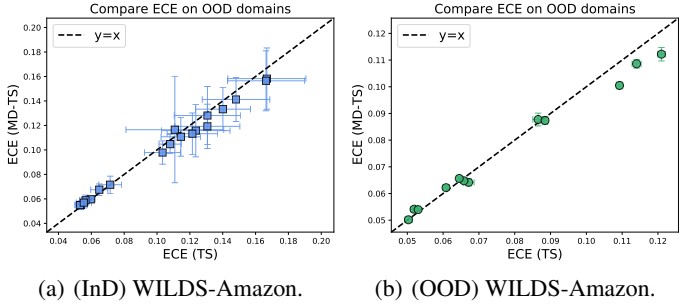

(a) (InD) WILDS-Amazon.  (b) (OOD) WILDS-Amazon.

Figure 7: (*Evaluated on WILDS-Amazon.*) Per-domain ECE of MD-TS and TS on both in-distribution domains and out-of-distribution domains. Each plot is shown with ECE of TS ($X$-axis) and ECE of MD-TS ($Y$-axis). Top: per-domain ECE evaluated on InD domains. Bottom: per-domain ECE evaluated on OOD domains. Lower ECE is better.

## C.8 Additional results on efficiency of MD-TS

**Efficiency.** We study the efficiency of our proposed method by measuring running time on three datasets (seconds). We consider the ResNet50 for all datasets. By using the standard `sklearn.linear_model`, it takes 39.8s/3.2s/4.1s on ImageNet-C/WILDS-RxRx1/GLDv2 for solving the linear regression problem of MD-TS, and the overall running time of MD-TS is 49.7s/4.6s/6.6s on ImageNet-C/WILDS-RxRx1/GLDv2. standard TS takes 7.8s/1.2s/3.6s on ImageNet-C/WILDS-RxRx1/GLDv2. We summarize the comparison results in Table 9, Appendix C.8.

Table 9: Compare the running time of MD-TS and TS on three datasets.

| Datasets | Architectures | RunTime (OLS of MD-TS) | Overall RunTime (MD-TS) | Overall RunTime (TS [Guo et al., 2017]) |
|---|---|---|---|---|
| ImageNet-C | ResNet-50 | 39.8s | 49.7s | 7.8s |
| WILDS-RxRx1 | ResNet-50 | 3.2s | 4.6s | 1.2s |
| GLDv2 | ResNet-50 | 4.1s | 6.6s | 3.6s |

## C.9 Additional experimental results on Brier Score

Table 10: Per-domain Brier Score [Brier et al., 1950] comparison on three datasets. We evaluate the per-domain Brier Score on InD and OOD domains. We report the mean and standard error of per-domain Brier Score on one dataset. Lower Brier Score means better performance.

| Datasets | Architectures | InD-domains | | | OOD-domains | | |
|---|---|---|---|---|---|---|---|
| | | MSP | TS | MD-TS | MSP | TS | MD-TS |
| ImageNet-C | ResNet-50 | 0.533±0.018 | 0.527±0.017 | 0.522±0.017 | 0.531±0.030 | 0.528±0.029 | **0.525±0.029** |
| | Efficientnet-b1 | 0.430±0.016 | 0.429±0.016 | 0.423±0.016 | 0.393±0.026 | 0.389±0.026 | **0.386±0.026** |
| | BiT-M-R50 | 0.475±0.017 | 0.475±0.017 | 0.465±0.016 | 0.447±0.030 | 0.448±0.029 | **0.441±0.029** |
| | ViT-Base | 0.327±0.014 | 0.327±0.014 | 0.324±0.014 | 0.268±0.019 | 0.268±0.019 | **0.267±0.019** |
| WILDS-RxRx1 | ResNet-50 | 1.053±0.019 | 0.908±0.009 | 0.890±0.010 | 0.904±0.010 | 0.841±0.005 | **0.813±0.007** |
| | ResNext-50 | 0.985±0.017 | 0.901±0.010 | 0.886±0.011 | 0.865±0.009 | 0.828±0.006 | **0.810±0.007** |
| | DenseNet-121 | 1.041±0.020 | 0.904±0.010 | 0.887±0.011 | 0.877±0.010 | 0.822±0.006 | **0.798±0.007** |
| GLDv2 | ResNet-50 | 0.788±0.003 | 0.787±0.002 | 0.782±0.003 | 0.792±0.005 | 0.792±0.004 | **0.791±0.004** |
| | BiT-M-R50 | 0.680±0.004 | 0.663±0.003 | 0.659±0.003 | 0.684±0.006 | 0.666±0.005 | **0.665 ±0.004** |
| | ViT-Small | 0.670±0.004 | 0.665±0.003 | 0.659±0.003 | 0.673±0.005 | 0.667±0.004 | **0.666±0.004** |

# D  Missing Proofs

In this section, we present the proof for Theorem 5.2.

**Theorem D.1** (Restatement of Theorem 5.2). *Let $\mathcal{H}$ be a hypothesis class that contains functions $h : \mathcal{X} \to [0,1]$ with pseudo-dimension $\mathrm{Pdim}(\mathcal{H}) = d$. Let $\{D_{k,X}\}_{k=1}^{K}$ denote the empirical distributions generated from $\{\mathcal{P}_{k,X}\}_{k=1}^{K}$, where $\mathcal{D}_{k,X}$ contains $n$ i.i.d. samples from the marginal feature distribution $\mathcal{P}_{k,X}$ of domain $k$. Then for $\delta \in (0,1)$, with probability at least $1 - \delta$, we have*

$$\varepsilon(\hat{h}, \widetilde{\mathcal{P}}_X) \leq \sum_{k=1}^{K} \hat{\alpha}_k \cdot \hat{\varepsilon}(\hat{h}, \mathcal{D}_{k,X}) + \frac{1}{2}\mathrm{d}_{\bar{\mathcal{H}}}(\mathcal{P}_{K,X}^{\hat{\alpha}}, \widetilde{\mathcal{P}}_X) + \lambda(\mathcal{P}_{K,X}^{\hat{\alpha}}, \widetilde{\mathcal{P}}_X) + \widetilde{O}\left(\frac{\mathrm{Pdim}(\mathcal{H})}{\sqrt{nK}}\right),$$

*where $\hat{\alpha}$ and $\lambda(\mathcal{P}_{K,X}^{\hat{\alpha}}, \widetilde{\mathcal{P}}_X)$ are defined in Eq. (3), $\widetilde{\mathcal{P}}_X$ denotes the marginal distribution of the OOD domain, $\mathrm{Pdim}(\mathcal{H})$ is the pseudo-dimension of the hypothesis class $\mathcal{H}$, and $\hat{\varepsilon}(\hat{h}, \mathcal{D}_{k,X})$ is the empirical risk of the hypothesis $\hat{h}$ on $\mathcal{D}_{k,X}$.*

Above, we use $\widetilde{O}(\cdot)$ to mean $O(\cdot)$ with some additional poly-logarithmic factors.

*Proof.* To start with, we use $\varepsilon(h, h', \mathcal{P}_X)$ to denote $\varepsilon(h, h', \mathcal{P}_X) = \mathbb{E}_{X \sim \mathcal{P}_X}[|h(X) - h'(X)|]$. Let $\varepsilon(\hat{h}, \widetilde{\mathcal{P}}_X)$ denote $\varepsilon(\hat{h}, \widetilde{h}^{\star}, \widetilde{\mathcal{P}}_X)$, where $\widetilde{h}^{\star}$ minimizes the risk under $\widetilde{\mathcal{P}}_X$. We can upper bound $\varepsilon(\hat{h}, \widetilde{h}^{\star}, \widetilde{\mathcal{P}}_X)$ by

$$\varepsilon(h, \widetilde{h}^{\star}, \widetilde{\mathcal{P}}_X) \leq \varepsilon(h^{\star}, \widetilde{h}^{\star}, \widetilde{\mathcal{P}}_X) + \varepsilon(h, h^{\star}, \widetilde{\mathcal{P}}_X),$$

by the triangle inequality (see Lemma 3 of Zhao et al. [2018]). Above, $h^{\star}$ is defined as

$$h^{\star} := \mathrm{argmin}_{h \in \mathcal{H}}\{\varepsilon(h, \mathcal{P}_{K,X}^{\alpha}) + \varepsilon(h, \widetilde{\mathcal{P}}_X)\}.$$

Next, we have

$$
\begin{aligned}
\varepsilon(h, \widetilde{h}^{\star}, \widetilde{\mathcal{P}}_X) &\leq \varepsilon(h^{\star}, \widetilde{h}^{\star}, \widetilde{\mathcal{P}}_X) + \varepsilon(h, h^{\star}, \widetilde{\mathcal{P}}_X) \\
&= \varepsilon(h^{\star}, \widetilde{h}^{\star}, \widetilde{\mathcal{P}}_X) + \varepsilon(h, h^{\star}, \widetilde{\mathcal{P}}_X) - \varepsilon(h, h^{\star}, \mathcal{P}_{K,X}^{\alpha}) + \varepsilon(h, h^{\star}, \mathcal{P}_{K,X}^{\alpha}) \\
&\leq \varepsilon(h^{\star}, \widetilde{h}^{\star}, \widetilde{\mathcal{P}}_X) + |\varepsilon(h, h^{\star}, \widetilde{\mathcal{P}}_X) - \varepsilon(h, h^{\star}, \mathcal{P}_{K,X}^{\alpha})| + \varepsilon(h, h^{\star}, \mathcal{P}_{K,X}^{\alpha}) \\
&\leq \varepsilon(h^{\star}, \widetilde{h}^{\star}, \widetilde{\mathcal{P}}_X) + \frac{1}{2}\mathrm{d}_{\bar{\mathcal{H}}}(\mathcal{P}_{K,X}^{\alpha}, \widetilde{\mathcal{P}}_X) + \varepsilon(h, h^{\star}, \mathcal{P}_{K,X}^{\alpha}),
\end{aligned}
$$

where the last step is using the Lemma 1 in Zhao et al. [2018]. Therefore, we have

$$
\begin{aligned}
&\varepsilon(h, \widetilde{h}^{\star}, \widetilde{\mathcal{P}}_X) \\
&\leq \varepsilon(h^{\star}, \widetilde{h}^{\star}, \widetilde{\mathcal{P}}_X) + \frac{1}{2}\mathrm{d}_{\bar{\mathcal{H}}}(\mathcal{P}_{K,X}^{\alpha}, \widetilde{\mathcal{P}}_X) + \varepsilon(h, h^{\star}, \mathcal{P}_{K,X}^{\alpha}) \\
&\leq \varepsilon(h^{\star}, \widetilde{h}^{\star}, \widetilde{\mathcal{P}}_X) + \frac{1}{2}\mathrm{d}_{\bar{\mathcal{H}}}(\mathcal{P}_{K,X}^{\alpha}, \widetilde{\mathcal{P}}_X) + \varepsilon(h, h_{K,\alpha}^{\star}, \mathcal{P}_{K,X}^{\alpha}) + \varepsilon(h^{\star}, h_{K,\alpha}^{\star}, \mathcal{P}_{K,X}^{\alpha}) \quad (5) \\
&= \varepsilon(h, h_{K,\alpha}^{\star}, \mathcal{P}_{K,X}^{\alpha}) + \frac{1}{2}\mathrm{d}_{\bar{\mathcal{H}}}(\mathcal{P}_{K,X}^{\alpha}, \widetilde{\mathcal{P}}_X) + \underbrace{\lambda(\mathcal{P}_{K,X}^{\alpha}, \widetilde{\mathcal{P}}_X)}_{:=\varepsilon(h^{\star}, \widetilde{h}^{\star}, \widetilde{\mathcal{P}}_X) + \varepsilon(h^{\star}, h_{K,\alpha}^{\star}, \mathcal{P}_{K,X}^{\alpha})},
\end{aligned}
$$

where we apply the triangle inequality in second step and $h_{K,\alpha}^{\star}$ minimizes the risk under $\mathcal{P}_{K,X}^{\alpha}$. Next, by Theorem 10.6 in Mohri et al. [2018], Theorem 1,2 and Lemma 2 in Zhao et al. [2018], we have for $\delta \in (0,1)$, with probability at least $1 - \delta$,

$$\varepsilon(h, h_{K,\alpha}^{\star}, \mathcal{P}_{K,X}^{\alpha}) \leq \hat{\varepsilon}(h, h_{K,\alpha}^{\star}, \mathcal{P}_{K,X}^{\alpha}) + \widetilde{O}\left(\frac{\mathrm{Pdim}(\mathcal{H})}{\sqrt{nK}}\right). \quad (6)$$

Meanwhile, based on its definition, $\hat{\varepsilon}(h, h_{K,\alpha}^{\star}, \mathcal{P}_{K,X}^{\alpha})$ can be rewritten as

$$\hat{\varepsilon}(h, h_{K,\alpha}^{\star}, \mathcal{P}_{K,X}^{\alpha}) = \sum_{k=1}^{K} \alpha_k \cdot \hat{\varepsilon}(h, h_k^{\star}, \mathcal{P}_{k,X}). \quad (7)$$

Putting Eq. (5), (6), and (7) together, we have for $\delta \in (0, 1)$, with probability at least $1 - \delta$,

$$\varepsilon(h, \widetilde{\mathcal{P}}_X) \leq \sum_{k=1}^{K} \alpha_k \cdot \hat{\varepsilon}(h, \mathcal{D}_{k,X}) + \frac{1}{2} d_{\bar{\mathcal{H}}}(\mathcal{P}_{K,X}^{\alpha}, \widetilde{\mathcal{P}}_X) + \lambda(\mathcal{P}_{K,X}^{\alpha}, \widetilde{\mathcal{P}}_X) + \widetilde{O}\left(\frac{\mathrm{Pdim}(\mathcal{H})}{\sqrt{nK}}\right)$$

hold for any $\alpha \in \Delta$ and $h \in \mathcal{H}$. Thus, we complete our proof.

$\square$