# OpenReview forum: "Robust Calibration with Multi-domain Temperature Scaling"
_NeurIPS.cc/2022/Conference — NeurIPS 2022 Accept_

### Official Review · Reviewer_xbg7 · 2022-06-24

**Rating:** 6
**Confidence:** 5
**Soundness:** 4 excellent
**Presentation:** 4 excellent
**Contribution:** 3 good

**Summary:**

This paper propose a multi-domain temperature scaling method for distribution calibration with distribution shifts. The heterogeneity of temperature is used to improve the robustness. Numerical experiments and theoretical results are presented to validate the method.

**Questions:**

Like TS, MD-TS should also adopt a validation dataset, right? Can we do multi-domain without the validation dataset?

**Ethics Review Area:**

["I don’t know"]

**Limitations:**

1. The baselines only include TS and max-softmax. Can you compare with the more recent method, such as node, sde-net and others?
2. How is the optimization goes? Is MD-TS still efficient, compared with TS?
3. In the introduction, only an example of multi-domain uncertainty? I suggest a more detailed elaboration on this motivation, since it’s the main contribution and claim.

**Strengths And Weaknesses:**

strengths:

1. well-motivated
2. clear presentation and good writing
3. extensive results to support

weakness

1. more rigorious  experimental results (more baselines and efficiency, detailed in the limitations)
2. more introduction to highlight the motivation

---

> ### Author Response · Authors · 2022-07-30
> **Clarification about recent methods (node, sde-net)**
>
> > Can you compare with the more recent method, such as node, sde-net and others?
>
> Thank you for suggesting these recent methods! Is this (link: http://proceedings.mlr.press/v119/kong20b/kong20b.pdf) the **sde-net** method you were referring to? Could you also clarify the reference for the **node** method? Thank you!

---

> ### Author Response · Authors · 2022-08-02
> **Response and thank you for your feedback (part 1)**
>
> We thank the reviewers for their careful reading of our paper and help with improving our manuscript. We sincerely appreciate that you find our work is '*well-motivated*' (**Reviewer xbg7**, **Reviewer 4JGV**), proposes '*a simple and effective*' method for calibration under distribution shift (**Reviewer oWGL**), conducts '*extensive/comprehensive*' experiments to support the findings (**Reviewer oWGL**, **Reviewer xbg7**), and '*provide interesting theoretical justification*' (**Reviewer xbg7**). Two big changes to the paper are the addition of new calibration baselines (MC dropout and deep ensembles) in Appendix B.7 and the inclusion of a new experiment on text data in Appendix B.8.
>
> In what follows, we try to address your concerns/questions and provide a detailed item-by-item response to your comments.
>
> ======================================================================================
>
> >**Q1**: *The baselines only include TS and max-softmax. Can you compare with the more recent method, such as node, sde-net and others?*
>
> **A1**: Thank you for your suggestions on experiments. Besides TS and max-softmax, we also compared our proposed approach with histogram binning (HistBin), isotonic regression (Isotonic), and Bayesian Binning into Quantiles (BBQ) in Table 5, Appendix B.4 in our initial submission. Meanwhile, we have conducted new experiments (in comparison to deep ensemble and MC dropout) as *Reviewer oWGL* suggested, and the new results have been added to Table 7, Appendix B.7 of our revised submission.
>
> Thank you for pointing out the SDE-Net [1] reference, we have added it to the related work section in our revised submission. We tried to implement SDE-Net on RxRx1 and GLDv2 datasets, and we have not managed to get satisfactory results due to the time limit. We only get less than 5% classification test accuracy on both datasets, whereas the networks considered in the paper achieve more than 35% test accuracy on both datasets. We think this is mainly due to the fact that: (1) SDE-Net requires network architecture changes and applies neural SDE models that are different from the standard models we considered in this paper; (2) We could not find SDE-Net models that are pre-trained on the ImageNet dataset, and we applied ImageNet pre-trained models for RxRx1 and GLDv2 in our experiments. We will add the comparison results (SDE-Net) in our final version.
>
> [1] SDE-Net: Equipping Deep Neural Networks with Uncertainty Estimates. Lingkai Kong, Jimeng Sun, Chao Zhang. Proceedings of the 37 th International Conference on Machine Learning, 2020.
>
> >**Q2**: *How is the optimization goes? Is MD-TS still efficient, compared with TS?*
>
> **A2**: Thank you for pointing this out. For the optimization part of MD-TS, it is computationally efficient to optimize the linear regression problem in MD-TS (in Step 2 of MD-TS). Meanwhile,  the Step 1 of MD-TS is running TS on smaller subsets.
> Overall, MD-TS is almost as efficient as TS. By using the standard sklearn.linear_model, it takes 39.8s/3.2s/4.1s on ImageNet-C/WILDS-RxRx1/GLDv2 for solving the linear regression problem of MD-TS, and the overall running time of MD-TS is 49.7s/4.6s/6.6s on ImageNet-C/WILDS-RxRx1/GLDv2. standard TS takes 7.8s/1.2s/3.6s on ImageNet-C/WILDS-RxRx1/GLDv2.  We have included the discussion on efficiency in Appendix B.9 and summarized the comparison results in Table 9, Appendix B.9 in our revised submission.
>
> >**Q3**: *In the introduction, only an example of multi-domain uncertainty? I suggest a more detailed elaboration on this motivation, since it’s the main contribution and claim.*
>
> **A3**: Thank you for your suggestion. We will add another two examples (listed below) of multi-domain uncertainty to the first paragraph of the introduction in our final version.
>
> '*As another example, a centralized model is trained on training data from existing clients in federated learning. It is important for the central server to provide uncertainty quantification for every client. Similar to the fMRI example, the centralized model should still produce valid uncertainty quantification for unseen new clients. Another example is applying animal recognition models on images in wildlife monitoring, where one set of camera traps corresponds to one domain, and the model will be deployed under distribution shift, i.e., new camera traps.*'

---

> ### Author Response · Authors · 2022-08-02
> **Response and thank you for your feedback (part 2)**
>
> >**Q4**: *Like TS, MD-TS should also adopt a validation dataset, right? Can we do multi-domain without the validation dataset?*
>
> **A4**: This is a great point and gets to the heart of MD-TS. The core reason that we are able to generalize better to new test domains is that the calibration is done with domains that are “fresh” to the model. That way, the uncertainty information is calibrated correctly for OOD tasks, so to speak. As you are suggesting, this is a critical aspect of our approach.
>
> Can we do this without the validation set? We have tried many other versions of MD-TS during development, and found that MD-TS with validation set had consistently better results than other variants. (We do not report on all the other less effective versions in the manuscript for lack of space, but we experimented with many of them in the course of this project.) Our current understanding is that the validation set approach gives the best uncertainty information for the reasons above. It may be possible to accomplish this without the validation set, and we will continue experimenting with this. Still, at present, the most robust way is to use a validation set.
>
>
> Lastly, we point out that even regular temperature scaling typically requires a validation set to avoid overfitting, so we are not adding any additional requirements above the usual temperature scaling approach.

---

> > ### Comment · Reviewer_xbg7 · 2022-08-03
> > **Thanks for the update.**
> >
> > Hi, authors, Thanks for your updates.
> >
> > * For 'node', I mean 'neural ordinary equations'.
> > * Thanks for your adding results for resolving my comments, which I think would help you improve this paper.
> > * As for the validation dataset, it must be lack of data on a targeted domain so it motivates you to do multi-domain. I disagree with 'your MD-TS is not adding additional requirements' beacuse MD-TS is applied on a new setting that lacks data on a new domain. Therefore, the validation dataset splitting might somehow limits your contribution of this paper. Still, I will remain my score as a weak accept. No matter this paper is accepted or not,  I suggest adding a discussion on this validation dataset splitting, which involves how the MD-TS performs without a validation dataset.

---

> > > ### Author Response · Authors · 2022-08-03
> > > **Thanks for the feedback**
> > >
> > > Thank you for engaging with us and helping us improve the paper.
> > >
> > > Thank you for your suggestion on the discussion of the validation dataset splitting. We will add a discussion of this point to the main text of our final version.

---

### Official Review · Reviewer_4JGV · 2022-07-11

**Rating:** 7
**Confidence:** 4
**Soundness:** 3 good
**Presentation:** 3 good
**Contribution:** 2 fair

**Summary:**

This paper proposes a type of temperature scaling that is more robust on out-of-distribution dataset. The work first learns K temperatures for K in-domain datasets. Then a linear regressor is learned to predict the K temperatures given feature embeddings, hence learning to predict a suitable temperature given a sample. The linear regressor is then applied to out-of-distribution samples, calibrating a model’s output. The paper verifies the idea on three datasets and illustrates improved calibration results by the proposed method.

**Questions:**

Questions
1.	In Definition 2.2, is the number of pooled data identical in each domain? If not, taking the average should not be enough. It should reflect the number of examples in each domain as well. Hence, taking expectation over K domains should be modified.


**Limitations:**

-	The concept of the proposed method is tested on image domain. It would be interesting to see the method on other domains, such as NLG (translation).

**Strengths And Weaknesses:**

Strength
-	This paper asks an important question: How should one choose temperature for predictions from out-of-distribution inputs.
-	The empirical result shows improved calibration score with the proposed method over the baselines.

Weakness
-	The assumption in the Theorem 5.2 is strong. The assumption starts by assuming that Out of distribution is similar to a mixture of in-domain distributions. This assumption, however, is too strong without proper citations or empirical results. If this assumption holds, then simply learning a universal temperature with data collected from the in-domains may be enough.

---

> ### Author Response · Authors · 2022-08-02
> **Response and thank you for your feedback (part 1)**
>
> We thank the reviewers for their careful reading of our paper and help with improving our manuscript. We sincerely appreciate that you find our work is '*well-motivated*' (**Reviewer xbg7**, **Reviewer 4JGV**), proposes '*a simple and effective*' method for calibration under distribution shift (**Reviewer oWGL**), conducts '*extensive/comprehensive*' experiments to support the findings (**Reviewer oWGL**, **Reviewer xbg7**), and '*provide interesting theoretical justification*' (**Reviewer xbg7**). Two big changes to the paper are the addition of new calibration baselines (MC dropout and deep ensembles) in Appendix B.7 and the inclusion of a new experiment on text data in Appendix B.8.
>
> In what follows, we try to address your concerns/questions and provide a detailed item-by-item response to your comments.
>
> ======================================================================================
>
> >**Q1**: *The assumption in the Theorem 5.2 is strong. The assumption starts by assuming that Out of distribution is similar to a mixture of in-domain distributions.*
>
> **A1**: Thank you for your valuable feedback. As you say, our upper bound in Eq. (4) is sharpest  when the OOD domain is similar to a mixture of in-domain distributions. Still, the theorem has informative content even when this doesn’t hold. In particular, even if the OOD domain is very different from the in-distribution domains, the Eq. (4) still implies that we could decrease the risk upper bound on the OOD domain if we perform multi-domain calibration. More specifically, if we could achieve good calibration performance on each individual domain by using multi-domain calibration (which that the first term in the RHS of Eq. (4) is small), then the second term plus the thrid term of RHS is always smaller or equal to $\frac{1}{2}d_{\bar{\mathscr{H}}}(\mathscr{P}^{\prime}, \widetilde{\mathscr{P}_X}) + \lambda(\mathscr{P}^{\prime}, \widetilde{\mathscr{P}_X})$, where $\mathscr{P}^{\prime}$ is the pooled distribution or any individual domain distribution. We have added a remark (Remark C.2 in Appendix C.2) to our revised submission for the better clarification.
>
> We also wish to provide more context for Theorem 5.2. We view this mainly as a sanity check, proving that our approach is theoretically sound in some particular case. This is complementary to our extensive experimental results that show that our approach is effective in practice. We view the experimental results as the heart of our paper, and we see that the method performs well even in cases where the assumptions of Theorem 5.2 likely do not hold.
>
> In summary, we partly agree with the reviewer’s criticism of Theorem 5.2, but wish to emphasize that we are not at all claiming that the assumptions are necessary for our method to succeed. Theorem 5.2 is one setting where theory can provide useful information about our method, but we observe good empirical performance much more broadly.
>
> >**Q2**: *This assumption, however, is too strong without proper citations or empirical results. If this assumption holds, then simply learning a universal temperature with data collected from the in-domains may be enough.*
>
> **A2**: Thank you for pointing this out. As suggested by Eq. (4) of Theorem 5.2, larger risks on in-distribution domains will lead to a larger upper bound for the risk evaluated on the OOD domain. On the other hand, as shown in Figure 1, a universal temperature is not sufficient to achieve good calibration performance on each individual in-distribution domain. In that experiment, there is a temperature that does well on the pooled data, but it does very poorly on when we evaluate it separately on its component domains. Therefore, even in the mixture of in-distribution domains setting, a universal temperature is suboptimal and applying multi-domain temperature scaling could be better than using a universal temperature. We have added a remark (Remark C.3 in Appendix C.2) to discuss this point explicitly in our revised manuscript. Thank you for surfacing this discussion.

---

> > ### Comment · Reviewer_4JGV · 2022-08-08
> > **Thank you for the detailed comments**
> >
> > I appreciate the additional work that the authors have done during the review session. All of the concerns have been dealt by the authors.

---

> > > ### Author Response · Authors · 2022-08-08
> > > **Thank you for the reply**
> > >
> > > We thank you again for your thoughtful review and valuable feedback!

---

> ### Author Response · Authors · 2022-08-02
> **Response and thank you for your feedback (part 2)**
>
> >**Q3**: *In Definition 2.2, is the number of pooled data identical in each domain? If not, taking the average should not be enough. It should reflect the number of examples in each domain as well. Hence, taking expectation over K domains should be modified.*
>
> **A3**: Thank you for pointing this out. The number of samples could be different across domains. In our definition, we weight each domain equally to balance across domains on purpose, which we hope  reflects how the calibration method performs on each individual domain. For example, if domain A has 100x the data as domain B, we still choose to weight them equally – our motivation for this choice is that we are hoping to achieve balanced coverage across many domains, to hopefully generalize to entirely new domains. Furthermore, in our experiments, we also visualize the ECE measured on each domain to provide additional information on model performance on every domain. We have added this discussion into Remark C.1 in Appendix C.1 to our revised submission to address this point to readers.
>
> >**Q4**: *The concept of the proposed method is tested on image domain. It would be interesting to see the method on other domains, such as NLG (translation).*
>
> **A4**: Thank you for your suggestion – this is very interesting to us. We have conducted new experiments on a NLP dataset – WILDS-Amazon (Amazon review dataset, multi-class sentiment classification task on review text, with a DistilBERT model). We added the new experimental results to Table 8 and Figure 9 in Appendix B.8 of revised submission. The results are exciting; we find that our proposed approach also outperforms existing methods on this NLP dataset. We think that the fundamental reason for the improved performance in the NLP experiment is again that using heterogeneous data leads to more robustness to new domains, and it is encouraging to see this play out in the empirical results. We will add more results (other models such as RoBERTa) on this dataset in the main text of our final version.

---

### Official Review · Reviewer_oWGL · 2022-07-13

**Rating:** 7
**Confidence:** 3
**Soundness:** 3 good
**Presentation:** 3 good
**Contribution:** 3 good

**Summary:**

The paper proposes a simple method for calibration that is robust to distributional shifts in the data. The proposed method extends the temperature scaling approach by learning to predict a suitable temperature value on an unseen domain.
The authors show empirically that their approach (MD-TS) can outperform other approaches (including vanilla temp scaling) on both in- and out-of-distribution test sets.
Along with this, the authors also provide a theoretical justification for their proposed method.

Calibration methods generally fail to remain robust on out-of-distribution examples and this paper works towards solving an important problem


**Questions:**

Why was the temperature prediction model chosen to be linear?
Prior work has experimented with methods to learn to predict a temperature value (https://arxiv.org/pdf/1903.00802.pdf): they use a 2 layer MLP to predict the temperature.

The ECE metric can be very sensitive when the number of examples in the test set are small and are not distributed over all bins. In such cases it might be interesting to also compare on other more standard metrics (apart from MAE) like the Brier Score.


**Limitations:**

The authors do not have a section detailing potential societal impacts, but they talk about how calibration can be useful in critical applications in the introduction.

**Strengths And Weaknesses:**

Strengths:
- The proposed method is both simple and effective at calibrating several models over multiple domains (an important problem) (+ originality, +quality + significance)
- The paper provides an interesting theoretical justification for their proposed approach which gives a better intuition for regarding calibration over multiple domains (+originality,  + quality, +significance)
- The analysis seems comprehensive; the authors also provide ablations for their approach
- The paper is well explained and easy to follow (+clarity)

Weaknesses:
- I felt that the chosen baselines were not very strong. [1] show that temperature scaling by itself isn't very robust on out-of-distr. examples. Perhaps the authors could compare against some of the methods shown in [1] to be more robust such as deep ensembles and MC dropout.


[1] Ovadia Y, Fertig E, Ren J, Nado Z, Sculley D, Nowozin S, Dillon J, Lakshminarayanan B, Snoek J. Can you trust your model's uncertainty? evaluating predictive uncertainty under dataset shift. Advances in neural information processing systems. 2019;32.


Overall, I feel this is a worthy contribution and vote for acceptance. Willing to increase my score if there's a more comprehensive comparison to previous methods.

---

> ### Author Response · Authors · 2022-08-02
> **Response and thank you for your feedback**
>
> We thank the reviewers for their careful reading of our paper and help with improving our manuscript. We sincerely appreciate that you find our work is '*well-motivated*' (**Reviewer xbg7**, **Reviewer 4JGV**), proposes '*a simple and effective*' method for calibration under distribution shift (**Reviewer oWGL**), conducts '*extensive/comprehensive*' experiments to support the findings (**Reviewer oWGL**, **Reviewer xbg7**), and '*provide interesting theoretical justification*' (**Reviewer xbg7**). Two big changes to the paper are the addition of new calibration baselines (MC dropout and deep ensembles) in Appendix B.7 and the inclusion of a new experiment on text data in Appendix B.8.
>
> In what follows, we try to address your concerns/questions and provide a detailed item-by-item response to your comments.
>
> ======================================================================================
>
> >**Q1**: *Compare against deep ensembles and MC dropout.*
>
> **A1**: Thank you for your suggestions on the experiments. We have conducted new experiments and compared our proposed approach to deep ensembles and MC dropout on both WILDS-RxRx1 and GLDv2. We have included these new results in Table 7, Appendix B.7 of our revised submission. We found that our proposed approach outperforms deep ensembles and MC dropout on both datasets. Also, in our revised submission, we summarize these results for the reader in the main text in Section 4.1 saying “Further comparisons in Appendix B.7 show that these improvements continue to hold when relative to two other calibration techniques: MC dropout and deep ensembles.”
>
> Due to computational constraints, we have not finished the experiments on ImageNet. We will add the comparison results (deep ensembles and MC dropout) on all datasets in the main text of our final version.
>
>
> >**Q2**: *Why was the temperature prediction model chosen to be linear?*
>
> **A2**: In our experiments, we found that simple linear regression, i.e., ordinary least squares (OLS), achieves competitive performance on all datasets we considered in this paper. We also investigated other approaches in Table 3, including non-parametric approaches such as kernel ridge regression (KRR) and K-nearest neighbors (KNN). We did not observe significant gains by using more complex approaches. Meanwhile, it is computationally efficient to solve linear regression and apply linear models for predicting temperature for new test samples.
>
>
> >**Q3**: *Prior work has experimented with methods to learn to predict a temperature value (https://arxiv.org/pdf/1903.00802.pdf): they use a 2 layer MLP to predict the temperature.*
>
> **A3**: Thank you for pointing out this reference. We have included this reference in the related work section of our revised submission. Also, we have conducted new experiments on using the 2-layer MLP to replace the linear model in our algorithm for predicting the temperature. Our preliminary results suggest that this 2-layer MLP does not outperform the linear model. For example, on WILDS-RxRx1 with ResNet50, the per-domain ECE of the 2-layer MLP is 5.62% whereas per-domain ECE of the linear model is 5.25%. We will include the results of 2-layer MLP in our final version.
>
>
> >**Q4**: *The ECE metric can be very sensitive when the number of examples in the test set are small and are not distributed over all bins. In such cases it might be interesting to also compare on other more standard metrics (apart from MAE) like the Brier Score.*
>
> **A4**: Thank you for your valuable suggestion. The alternative of the Brier score is a great idea, and it makes our evaluations more complete. We will include the Brier score results in our final version.
>
> As you say, we observe that the ECE requires a large number of data points, and we have been careful to find examples with enough data to reliably evaluate it. For the datasets we consider in this paper, most of the domains contain more than 2000 samples, where the ECE metric could provide reasonably good evaluations on different calibration methods.
>
>
> >**Q5**: *The authors do not have a section detailing potential societal impacts.*
>
> A5: Thank you for your suggestion. We have added the Societal Impact section (Appendix D) to our revised submission.

---

### Meta-Review · Area_Chair_TVSD · 2022-08-29

**Recommendation:** Accept
**Confidence:** Certain

**Metareview:**

Reviewers find the paper original, useful, thorough in its numerics (in the revision), and clearly written.

**Award:**

No

---

### Decision · Program_Chairs · 2022-09-14

Accept